## Evaluation of hypofractionated adaptive radiotherapy using the MR Linac in localised pancreatic cancer: protocol summary of the Emerald-Pancreas phase 1/expansion study located at Oxford University Hospital, UK

Suliana Teoh [1,2] Alexander Ooms [3] Ben George,[4] Rob Owens,[2,4] Kwun-Ye Chu,[1,2] Joe Drabble,[4] Maxwell Robinson,[2] Matthew J Parkes [5] Lynda Swan,[1] Lucinda Griffiths,[1] Killian Nugent [2,4] James Good,[4] Tim Maughan,[1,2] Somnath Mukherjee[1,2]

**Correspondence to**
Dr Somnath Mukherjee;
somnath.mukherjee@oncology.ox.ac.uk

## ABSTRACT

**Introduction** Online adaptive MR-guided radiotherapy allows for dose escalation to pancreatic cancer while sparing surrounding critical organs. We seek to evaluate the safety of delivering hypofractionated five-fraction, three-fraction and single-fraction MR-guided stereotactic ablative radiotherapy (SABR) to the pancreas.

**Methods and analysis** This is a single-centre three-arm phase 1 non-randomised safety study. Patients with localised pancreatic cancer will receive either 50 Gy in five (biological equivalent dose ($BED_{10}$=100 Gy), 39 Gy in three ($BED_{10}$=90 Gy) or 25 Gy in a single fraction ($BED_{10}$=87.5 Gy) MR-guided daily online adaptive radiotherapy. Each fractionation regimen will be assessed as independent cohorts to determine tolerability, assessed continuously using Bayesian conjugate posterior beta distributions. The primary endpoint of the study is to establish the safety of five-fraction, three-fraction and single-fraction MR-guided hypofractionation SABR in localised pancreatic cancer by assessing dose-limiting toxicities. Secondary endpoints include overall survival, progression-free survival, local control rates, overall control rate, resection rates, long-term toxicities and freedom from second-line chemotherapy. This study plans to also explore imaging and immune biomarkers that may be useful to predict outcome and personalise treatment. The trial will recruit up to 60 patients with a safety run-in.

**Ethics and dissemination** The trial is approved by the West Midlands—Black Country Research Ethics Committee 22/WM/0122. The results will be disseminated via conference presentations, peer-reviewed scientific journals and submission to regulatory authorities. The data collected for the study, including individual participant data, will be made available to researchers on request to the study team and with appropriate reason, via octo-enquiries@oncology.ox.ac.uk. The shared data will be deidentified participant data and will be available for 3 years following publication of the study. Data will be shared with investigator support,

### STRENGTHS AND LIMITATIONS OF THIS STUDY

⇒ This is a phase I safety study evaluating online adaptive extreme hypofractionated radiotherapy using novel MR-guided-radiotherapy (strength).

⇒ There is continuous monitoring of the primary outcome of dose-limiting toxicity using conjugate posterior beta distribution (strength).

⇒ Longer-term outcome data are limited due to follow-up period of up to 24 months (limitation).

⇒ The trial has a non-comparative design (limitation).

⇒ All three fractionation regimens will be considered as three separate cohorts, and therefore, outcome could be that three different stereotactic ablative radiotherapy prescriptions are safe to deliver in pancreatic cancer (strength).

after approval of a proposal and with a signed data access agreement.

**Trial registration number** ISRCTN10557832

## BACKGROUND

The role of radiotherapy (RT) in patients with inoperable pancreatic cancer is controversial due to its high propensity to metastasise. Meaningful benefit of local control is challenging to quantify when most patients die of distant metastatic disease. Nevertheless, not all patients with adenocarcinoma of the pancreas develop distant metastatic disease. One-third of these patients die mainly of local progression.[1] As systemic treatment improves, local control is likely to be increasingly important. Stereotactic ablative radiotherapy (SABR) where an ablative dose of RT is delivered to a small volume in 1–5 fractions,

has been shown to achieve local control rates as high as 80%–100%,[2–8] compared with about 50%–70% reported in chemoradiotherapy (CRT) trials.[9 10] Initial SABR experiences for treatment of locally advanced pancreatic cancer with ablative doses of radiation had reported high rates of toxicity, particularly with regimens delivered in less than five fractions.[5 6 11]

There are a number of challenges in delivering highly conformal RT in pancreatic cancer. First, tumour and organs-at-risk (OAR) visualisation during treatment setup is limited with CT-based imaging. Moreover, the proximity of pancreatic tumours to highly mobile radiosensitive organs such as the duodenum, stomach and bowel means delivering high doses while sparing these organs is difficult. A number of studies have explored the use of SABR to improve outcome. Of concern is the toxicity rate observed in some of these studies. Hoyer *et al* treated patients with 45 Gy in 3 fractions and reported 5 out of 22 patients developing severe mucositis or perforation.[11] Not all patients had on-board CT-imaging for RT setup. Similarly, Liauw *et al* reported up to 27% grade 3 and above late toxicities following three-fraction SABR.[6] In both cases, large planning target volume (PTV) margin and no motion management was used. Schellenberg *et al* evaluated concurrent gemcitabine with single fraction 25 Gy SABR for LAPC and reported freedom from local progression was 88% at 1 year. However, they reported 15% grade 2 ulcers and one patient with a duodenal perforation.[12]

Daily online adaptive MR-guided radiotherapy (MRgRT) is a novel method of delivering RT that could potentially allow for dose escalation without exceeding dose to the OAR. MRI provides superior visualisation of organs such as the stomach, duodenum and bowel as well as improved tumour identification. Online adaptive MRgRT gives the opportunity to adapt treatment plans daily to account for interfraction variability of these organs to maintain target coverage while sparing high doses to OAR.[13] Furthermore, the ability to track the tumour during treatment delivery means 'real-time' intrafraction monitoring is possible.

Early data from a retrospective study show that dose escalation to 50 Gy in five fractions using online adaptive MRgRT is safe and may result in improved survival.[14] This retrospective analysis of 42 locally advanced pancreatic cancer patients treated by MR-guided adaptive RT at four institutions demonstrated that high-dose SABR or hypofractionated radiation therapy delivered using daily adaptive dose planning has the potential to further improve overall survival.[14] A control group of 19 patients treated to more conventional radiation doses without frequent dose adaptation showed a median survival of 14.8 months, while patients treated to high radiation doses (n=23, maximum BED10 of >90 Gy) under daily or almost daily adaptive replanning had an estimated median survival of 27.8 months (p=0.005). Increased radiation dose delivery using daily dose adaptation was correlated with less grade 3 toxicity (0% in the high dose group vs 15.8% in patients

treated to lower radiation doses without dose adaptation). Hall *et al* summarised the current published experience for MRgRT in pancreatic cancer and found local control rates of between 77% and 88% at 1 year with toxicity rates of less than 10% (n=141 from 6 studies mostly retrospective).[15] Chuong *et al* retrospectively analysed 50 patients with pancreatic cancer treated with 50 Gy in 5 fraction using online daily adaptation on the ViewRay MRlinac and found 1-year and 2-year estimated local control were 97.8% and 88.9%, respectively. Median survival was 21 months (1-year and 2-year OS 87.9% and 50%) (European Society for Radiotherapy and Oncology (ESTRO) 2021).[16]

Pancreatic adenocarcinoma features both local and systemic immunosuppression.[17] This enables immune escape and establishment of the tumour both at the primary and distant sites. Whether or not ablative radiation doses to the primary tumour can induce systemic tumour immunity in pancreatic cancer is unknown. In murine pancreatic ductal adenocarcinoma (PDAC) models, SABR has been shown to induce immunogenic cell death (ICD) and promote tumour cell antigen presentation resulting in activation of tumour-specific T cells.[18] Mills *et al* evaluated the immune response from histopathological samples of patients treated with SABR (25 Gy in 5 fractions) preoperatively versus surgery alone and found that SABR reduced PDAC cell density, induced ICD and increased the proportion of PD-1+T cells. However, despite these findings, intratumour T cell levels remain low and outnumbered by myeloid suppressors.[19] While SABR to an unselected population of LAPC patients is unlikely to significantly impact survival outcomes, there is potential for a multimodality approach with immunotherapy, particularly when combined with biomarkers suggestive of a locally aggressive rather than metastatic phenotype.

The Emerald-Pancreas trial will evaluate the safety of delivering five, three and single fractions SABR to the pancreas using online adaptive MRgRT. Shorter fractionation schedules are more convenient for patients and increase access to and cost-effectiveness of the scarce and expensive resource of an MRgRT machines. Being able to achieve high rates of local control of the primary tumour with minimal side effects and limited impact on patient lives would be a great advantage for patients. It will also enable greater access for systemic therapies, including conventional chemotherapy and novel immunotherapies, to develop more effective long-term control regimens to combat this highly aggressive disease.

## METHODS/DESIGN

The aim of the trial is to assess safety of extreme hypofractionation of SABR using MRgRT in pancreatic cancer. The five-fraction, three-fraction and single-fraction MRgRT treatments will be assessed as independent cohorts to determine if each one is tolerable. Tolerability will be assessed using conjugate posterior beta distributions.

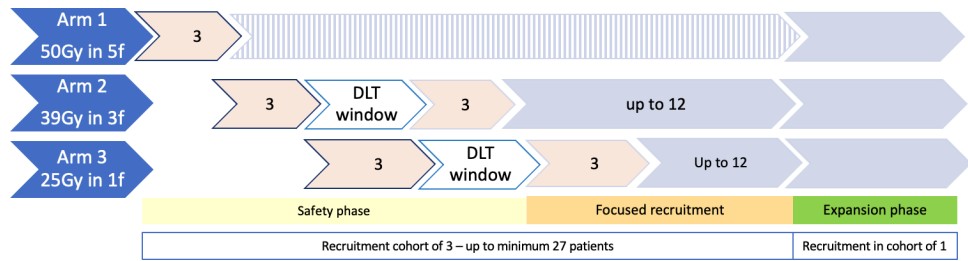

**Figure 1** Trial recruitment phases. DLT, dose-limiting toxicities.

## PATIENT AND PUBLIC INVOLVEMENT

The Emerald trial has been reviewed and endorsed by patient and carer representatives. Patient and public involvement (PPI) began at the protocol design and development stage. This included a review of the trial patient information sheet and consent form. Further consumer feedback was obtained at the CTRad proposal guidance meeting. Their feedback was adopted and incorporated into the final version of the patient information sheet. PPI representatives will also be invited to attend trial management group meetings throughout the trial.

## TRIAL DESIGN

This is a three-arm, non-randomised, open, safety study. There are three phases of recruitment to the study: an initial safety run-in, a focused recruitment phase and an expansion phase (see figure 1).

In the initial safety run-in, three patients will be recruited into the five-fraction regimen. The three-fraction regimen will open immediately once three patients have been recruited into a five-fraction regimen. Once three patients have been accrued to the three-fraction regimen, the single fraction regimen can open immediately. Recruitment to the one-fraction and three-fraction regimens will pause while waiting for 3 months dose-limiting toxicity (DLT) follow-up from the first three patients in each cohort. It is possible both the single-fraction and three-fraction regimens will be paused at the same time. Any patients recruited while both one-fraction and three-fraction regimens are paused will be assigned the five-fraction regimen.

Following the safety run-in, the focused recruitment will start. Recruitment will continue in cohorts of three alternating between the three-fraction and one-fraction regimens until a total of 12 patients in the one-fraction and three-fraction regimens have been recruited. Recruitment to this phase will be continuous. After 3 patients' full DLT follow-up data in any regimen, the beta binomial model for that regimen can be run to assess the safety stopping rule in the event of a DLT.

For the remainder of the trial, the expansion phase recruitment will be in cohorts of one to all three regimens and recruitment will be continuous, alternating between one, three and five fractions. Where plans that meet dose constraints are unable to be generated for single-fraction regimens that the patient will be assigned to the three-fraction regimen or alternatively the five-fraction regimen if recruitment to the three-fraction cohort is paused or closed. Where plans that meet dose constraints are unable to be generated for three-fraction regimens, that patient will be assigned to the five-fraction regimen. These patients will be included in the main analysis for the regimen they are treated in. A sensitivity analysis will be performed excluding these patients as they may be at higher risk of toxicity. Figure 2 shows the trial schema and overview of follow-up. The duration of the study will be up to 24 months from the patient starting RT. All participants will be followed up for at least 3 months and to the maximum time until study closure. The trial has been open to recruitment since August 2022 with the first patient first visit being on 25 August 2022 with the end of study date being the 31 December 2024. Our last patient first visit is planned for the 30 June 2024.

## TRIAL INTERVENTION

Patients will receive RT online adaptive MRgRT on Viewray MRIdian MR Linac.[20] True Fast Imaging with Steady State Precession T2-weighted/T1-weighted sequences are used to provide the cine MRI (real-time imaging) for beam gating, and volumetric imaging for OARs. SABR fractions will be delivered ideally on alternate days, treating at least twice per week. Daily treatment is permissible with at least 18 hours between fractions for five-fraction regimen.

## HYPOFRACTIONATION

For each of the three fractionations (50 Gy in five fractions, 39 Gy in three fractions or 25 Gy in a single fraction), we have defined a single dose level (see table 1). The starting regimen will be 50 Gy in 5 fractions (BED10=100 Gy). Our in-silico planning study showed that three and single-fraction pancreas SABR plans could be generated while meeting organ dose constraints and delivering a meaningful dose to the target. Furthermore, treatments could be delivered within a reasonable time frame.[21] Patients included in the study were from the CAP cohort (n=8, median gross tumour volume (GTV) 41.35cc (range 15.9–64.4). The median PTV V100 coverage for 39 Gy in three fractions and 25 Gy in one fraction was 75.7% (60.6%–91.6%) and 66.1 (60.1%–84.2%), respectively. The median treatment delivery times for 15.2 min

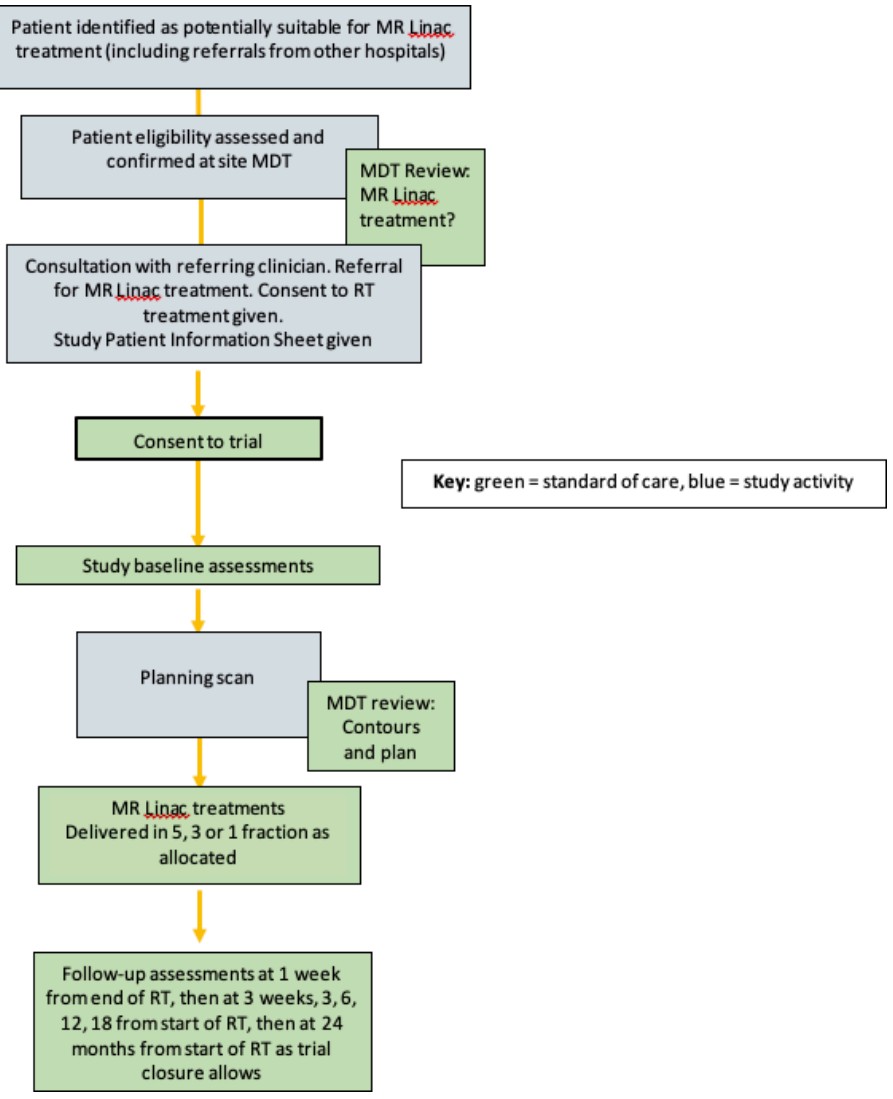

**Figure 2** Trial schema and overview of follow-up. MDT, multidisciplinary team; RT, radiotherapy.

(12.5–21.7 min) and 21.0 min (15.9–33.2 min) for 39 Gy in three fractions and 25 Gy in a single fraction, respectively.

## DOSE-LIMITING TOXICITY

In this study, DLT is defined in the following list of possible SABR treatment-related adverse events (defined according to the Common Terminology

| Table 1 | Radiotherapy dose levels | | |
|---|---|---|---|
| | **Dose to PTV (Gy)** | | |
| **Fractions** | **Dose/#** | **Total dose** | **BED 10** |
| 5 | 10 | 50 | 100 |
| 3 | 13 | 39 | 90 |
| 1 | 25 | 25 | 88 |

BED 10 biological equivalent dose for acute reacting tissues ($\alpha/\beta=10$).
BED, biological equivalent dose; PTV, planning target volume.

Criteria for Advserse Events (CTCAE V.5.0) seen in the period from starting SABR treatment to 3 months post-treatment.

DLT events (within 3 months)
► Grade 3 upper gastro-intestinal (GI) bleeding.
► GI fistula (any grade).
► Grade 4 nausea/vomiting uncontrolled despite optimum antiemetics.
► Grade 4 pancreatitis not stent related.
► Vascular events (where these are not considered to be tumour related).

## LATE-ONSET SEVERE TOXICITIES

Late-onset severe toxicities may occur and will be monitored for during follow-up (>3 months and up to 24 months where trial remains open and/or patient on trial).
► ≥Grade 3 upper GI bleed.
► GI fistula (any grade).
► ≥Grade 3 vascular events.

## TRIAL POPULATION

The target population for the Emerald-Pancreas trial consists of male or female aged 16 years or above with an ECOG performance status of 0–1 with a localised pancreatic cancer. Patients are not permitted to enter the trial if they have a contraindication to having an MRI (presence of metallic implants, shrapnel, claustrophobia or other expected intolerance of prolonged stay in an MRI scanner). The trial is based at a single UK centre (Oxford University Hospital NHS Foundation Trust).

There are no specific restrictions on tumour size, number or interval from diagnosis. The following patients with localised pancreatic cancer may enter the trial: locally advanced and inoperable pancreatic cancer, inoperable on medical grounds, operable, but declines surgery and locally recurrent pancreatic cancer.

A full list of inclusion and exclusion criteria can be found in online supplemental appendix 1. At least 3 months of chemotherapy prior to RT is recommended but not mandated. Chemotherapy should be avoided for at least 2 weeks before and for 4 weeks after RT.

## STUDY OBJECTIVES AND ENDPOINTS

The primary objective of the study is to establish the safety of five-fraction, three-fraction and single-fraction MR-guided hypofractionation SABR in localised pancreatic cancer by assessing DLTs up to 3 months from start of RT. The secondary objectives include assessment of efficacy and long-term toxicity with the following endpoints: overall survival, progression-free survival, local control rates, overall control rate, resection rates, long-term toxicities and freedom from second line chemotherapy.

Planned Exploratory objectives for Emerald-Pancreas trial will include evaluation of imaging and immune biomarkers that may be useful to predict outcome from RT. Research blood samples will be collected baseline, 3 weeks and 3 months after the first RT fraction to monitor changes in immune cell markers in response to treatment in a cohort of up to 20 patients. The schedule for collection of data for the purposes of evaluating all trial endpoints is found in online supplemental appendix 2.

## RT PLANNING

Patients will undergo a dedicated MR-Linac simulation as well as a CT simulation. CT simulation is to be used as primary data set for electron density information for planning dose calculation purposes. Alternatively, creation of synthetic CT simulation is acceptable where a suitable review of dosimetric accuracy using this approach has been carried out. The patient will be fasted 4 hours prior to scan and treatment. Intravenous contrast is required for the planning CT but not the planning MRI. The patient will be scanned in breath hold. A suitable tracking structure will be identified, either tumour in the first instance or suitable surrogate where tumour tracking is

determined unsuitable by clinical team. Ability to track motion will be assessed through MR cine imaging.

The planning CT scan should ideally be carried out post MR simulation. Scan will be in the same breath hold used during MR simulation to maximise the chance of accurate deformation of the CT for use in planning calculations.

## DELINEATION

Tumour volume definition is discussed and peer-reviewed with a pancreatic/upper GI radiologist and a second upper GI clinical oncologist. Outlining of the volumes of interest should take account of all available diagnostic imaging. GTV should not be reduced for a region that is negative on PET-CT but identified as abnormal on other imaging modalities. The following structures are to be created:

GTV_T=Macroscopic pancreatic tumour visible on imaging.

GTV_N=peritumoural lymph nodes >1 cm in short axis diameter and likely involved on diagnostic imaging. Include nodes <1 cm if considered involved by a radiologist. Prophylactic nodal irradiation is not typically performed.

GTV_Final=GTV_T+GTV_N.

Clinical target volume (CTV)=GTV+2 mm, with CTV cropped to the visceral OARs.

PTV=CTV+3 mm.

The PTV will be divided into PTV_high and PTV_low so that any overlap of OAR and PTV still allows tolerance of OAR to be achieved. The overlap region needs to be designed and set up as rule so that it can be updated during plan adaptation. Nomenclature is as follows:

Critical structures (CS)=duodenum+small bowel+large bowel+stomach.

PTV_high=PTV–(CS+5 mm*).

PTV_low=intersection between PTV and (CS+5 mm*).

*A 5–8 mm margin may be needed from critical structures to achieve plan objectives.

Organs at risk defined below are to be contoured, plus any others considered relevant for the particular case as deemed by CI:

► Duodenum, stomach, small bowel, large bowel and oesophagus:
   Should be outlined on all slices from 3 cm above to 3 cm below the PTV. Individual loops of duodenum, small bowel and large bowel should be outlined.
► Kidneys.
► Liver.
► Spinal cord.
► Common bile duct, vessels and gallbladder are to be contoured when near to or overlapping the PTV to avoid hotspots.

## RT PLANNING

Coplanar step-and-shoot IMRT plans are generated in a pseudoarc formation. The planning objective is that 98%

**Table 2** Organs-at-risk dose constraints

| Organ | Constraint | Volume |
|---|---|---|
| **Five fractions** | | |
| Unified viscera constraint (Stomach Duodenum Small bowel Large bowel) | V36 | 0.1 cc |
| | V33 | 0.5 cc |
| | V25 | 20 cc |
| Common bile duct | V50 | 0.5 cc |
| Liver | Mean | 13 Gy |
| Kidney | Mean to either | 10 Gy |
| Spinal cord PRV | V25 | 0.5 cc |
| | V14.5 | 1 cc |
| Aorta/IVC | V53 | 0.5 cc |
| **Three fractions** | | |
| Stomach Duodenum Small bowel Large bowel | V24 | 0.5 cc (mandatory) |
| | V22.2 | 0.5 cc (optimal) |
| | V20 | 5 cc (mandatory) |
| | V16.5 | 5 cc (optimal) |
| Common bile duct | V50 | 0.5 cc |
| Liver | V15 | 700 cc |
| Oesophagus | V25 | 0.5 cc |
| Kidneys | V16 | 200 cc |
| Spinal cord PRV | V18 | 0.1 cc |
| Aorta/IVC | V45 | 0.5 cc |
| **One fraction** | | |
| Duodenum Stomach Small bowel Large bowel | V12 | 0.5 cc (optimal) |
| | V15 | 0.5 cc (mandatory) |
| | V11 | 5 cc |
| Common bile duct | V27 | 0.5 cc |
| Liver | V9 | 700 cc |
| Oesophagus | V19 | 0.5 cc |
| Kidneys | V16 | 200 cc |
| Spinal cord PRV | V14 | 0.1 cc |
| Aorta/IVC | V37 | 0.5 cc |

All constraints are mandatory unless otherwise stated.
IVC, inferior vena cava; PRV, planning organ at risk volume; Vx, volume of organ receiving xGy.

of PTV_high receives ≥95% of the prescribed dose. If the mandatory duodenum, bowel or stomach constraints cannot be met, PTV coverage should be reduced until the constraints are met. PTV_high coverage should not ideally need compromising, but coverage should also be reduced until mandatory constraints are met where needed. The minimum dose coverage objective of PTV V100% ≥60%. The maximum dose in 1 cc of the PTV will be ≥125% and ≤ 140%. Hotspots should be within the GTV. OAR constraints are found in table 2. All constraints are mandatory unless otherwise stated.

## TREATMENT DELIVERY

System integrated image registration between the simulation image dataset and the fraction image dataset will be performed. Original plan contours are propagated onto the respective fraction image dataset. All critical structures within a 3 cm axial and 2 cm craniocaudal distance from the surface of the original PTV will be recontoured on the fraction image dataset. Tumour volume to be recontoured at clinician discretion.

An estimated delivered dose will be calculated using the software on the console (dose prediction). Subsequently, an adapted RT plan is generated. During radiation dose delivery, continuous cine MRI acquisition in at least one principal plane (suggested sagittal, but at the discretion of the treating physician) is mandatory for soft tissue tracking and radiation beam gating. To this end, a tracking slice will be positioned to include a cross-sectional cut of the target or suitable surrogate for intrafractional soft tissue tracking. The tracking/gating volume will be delineated based on either the GTV or the PTV.

Breathing motion management will be employed. This will include shallow breathing and breath hold. Breath hold may be patient directed or based on staff coaching. Breath-hold assistance devices such as use of mirrors to visualise a wall mounted monitor, MR compatible goggles or image projection into the bore for target positional visualisation are allowable and encouraged for use.

In patients receiving single fraction, the fraction will be subdivided into two pseudofractions to allow for a second IGRT match to be carried out. If this match is found to be acceptable and the patient is tolerating the treatment, the rest of the fraction will be delivered immediately. Otherwise, a second adaptive plan could be generated.

## PRE-RT CHECKS

Prospective contour and plan review will be done with an upper-GI radiologist and a second upper-GI site specialist. During treatment, attempts will be made to ensure a second MRL clinician is present. On treatment, secondary quality assurance (QA) contour check will be performed by a trained dosimetrist/medical physicist.

All complex plans must undergo patient specific pretreatment or on-treatment QA according to local protocols with a fluence and point dose delivery check on the linac recommended using a suitable QA detector array device (eg, ArcCheck). Pretreatment QA measurements must use the baseline treatment plan. On-treatment QA should use the most recently delivered treatment fraction.

## ANALYSIS PLAN

A maximum of 60 evaluable patients will be included. There is no formal power calculation as this is a phase I trial, but simulations have been carried out to ensure the sample size is adequate. Conjugate posterior beta distributions will be used to evaluate the safety of five, three and single fraction using established dose constraints.

The three fractionation regimens will be considered as three separate cohorts and analysed independently. The acceptable toxicity level is defined as 15% for each regimen. If the posterior probability of the DLT rate being above 15% is too high, (eg, P(risk of DLT >0.15 | Regimen, Data) > δ with δ defined using simulations before the start of the trial), that regimen will stop for safety. There is no early stopping for success.

A prior beta distribution will be specified for each regimen's toxicity rate. As binary (0=no DLT, 1=DLT) toxicity data are accrued in each regimen, these priors will be updated to conjugate posterior beta distributions. These represent the distribution for the probability of DLT rate within a specific regimen based on all available data. It is from these posterior distributions that inferences about safety will be made. The priors are calibrated to ensure the models provide sensible posterior probabilities based on prior clinical knowledge and incoming trial data.

Only patients for whom we have full information (eg, experienced a DLT or completed DLT follow-up window) will be included in the modelling. Due to the low acceptable toxicity rate (15%), there is a concern a small number of DLTs early in the trial could result in the trial stopping for safety when in reality the treatment is safe. To reduce the possibility of erroneously stopping early, the model for each regimen will not run until three patients from that regimen have full toxicity information. If there are safety concerns prior to this, the trial management group may convene and decide the appropriate action.

## Ethics and dissemination

The trial is approved by the West Midlands— Black Country Research Ethics Committee 22/WM/0122. The results will be disseminated via conference presentations, peer-reviewed scientific journals and submission to regulatory authorities. The data collected for the study, including individual participant data, will be made available to researchers on request to the study team and with appropriate reason, via octo-enquiries@oncology.ox.ac.uk. The shared data will be deidentified participant data and will be available for 3 years following publication of the study. Data will be shared with investigator support, after approval of a proposal and with a signed data access agreement.

**Author affiliations**
¹Department of Oncology, University of Oxford, Oxford, UK
²Department of Oncology, Oxford University Hospitals NHS Foundation Trust, Oxford, UK
³Nuffield Department of Orthopaedics, Rheumatology and Musculoskeletal Disorders, University of Oxford, Oxford, UK
⁴Department of Stereotactic and MR-guided Radiotherapy, GenesisCare UK, Oxford, UK
⁵Oxford Clinical Trials Research Unit (OCTRU), Oxford University, Oxford, UK

**Acknowledgements** The trial is coordinated by the Oncology Clinical Trials Office (OCTO) at the University of Oxford with statistical support from the Centre for Statistics in Medicine, which together form part of the Oxford Clinical Trials Research Unit (OCTRU), a UKCRC Registered Clinical Trials Unit.

**Contributors** All contributors meet at least one of the criteria recommended by the ICMJE. ST, TM and SM conceived the study design. ST, AO, BG, RO, K-YC, JD, MR, MJP, LS, LG, KN, JG, TM and SM were involved in protocol development and manuscript.

**Funding** This work is funded through the Oxford University-Genesis Care Collaboration Fund (award/grant number: n/a), the John Black Charitable Foundation (award/grant number: n/a) and Oxford Institute for Radiation Oncology, University of Oxford (award/grant number: n/a). ST is Clinical Career Development Fellow funded by CRUK (award:ART-Net grant). SM is part-funded by Biomedical Research Oxford (award/grant number: n/a). Standard treatment costs for radiotherapy are funded by the NHS (award/grant number: n/a).

**Competing interests** None declared.

**Patient and public involvement** Patients and/or the public were involved in the design, or conduct, or reporting, or dissemination plans of this research. Refer to the Methods section for further details.

**Patient consent for publication** Not applicable.

**Provenance and peer review** Not commissioned; externally peer reviewed.

**ORCID iDs**
Suliana Teoh http://orcid.org/0000-0002-5456-9273
Alexander Ooms http://orcid.org/0000-0003-1877-8323
Matthew J Parkes http://orcid.org/0000-0002-1574-9933
Killian Nugent http://orcid.org/0000-0001-6225-9497

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
