## [Reviewer comments · BMJ Open]

ARTICLE DETAILS

TITLE (PROVISIONAL)	Evaluation of hypofractionated adaptive radiotherapy using the MR Linac in localised pancreatic cancer: Protocol summary of the Emerald-Pancreas Phase 1/expansion study located at Oxford University Hospital UK.
AUTHORS	Teoh, Suliana; Ooms, Alexander; George, Ben; Owens, Rob; Chu, Kwun-Ye; Drabble, Joe; Robinson, Maxwell; Parkes, Matthew; Swan, Lynda; Griffiths, Lucinda; Nugent, Killian; Good, James; Maughan, Tim; Mukherjee, Somnath

VERSION 1 – REVIEW

REVIEWER	Chuong, Michael D. Miami Cancer Institute
REVIEW RETURNED	04-Dec-2022

GENERAL COMMENTS	The authors present a phase I study design of ablative MR-guided radiation therapy for pancreas cancers that are not operable that includes 1-, 3-, and 5-fraction regimens. Overall the manuscript is well-written and addresses a topic that is of significant clinical interest to patients and the medical community. Please consider the following: -there is very little difference in absolute fraction numbers between 1, 3, and 5. It would be helpful for the reader to understand better the rationale for studying these 3 fractionation schedules as opposed to only 1 and 5 fractions for example. If all 3 regimens are deemed safe then do the investigators plan to proceed with other studies incorporating all 3 regimens?-a phase II study (NCT03621644) of 50 Gy in 5 fractions delivered on MRIdian for pancreas cancers was recently presented at ASTRO showing very little grade 3+ toxicity. Considering including mention of this study and citing it. Also, how do the authors view the significance of the current trial that also is evaluating 50 Gy in 5 fractions on MRIdian in the context of the favorable results from the aforementioned study?-although this is a phase I study primarily evaluating toxicity, the follow-up period is relatively lengthy at over 1 year with secondary endpoints evaluating efficacy which seems more consistent with a phase II or III study design. Would it therefore be advantageous to be a bit more selective in patients enrolled based on CA19-9, receipt of prior chemotherapy, etc? Otherwise the long-term efficacy outcomes will be hard to interpret and as such shortening the follow-up period may be advantageous.-CA19-9 is assessed at baseline and then only at 2 of the follow-up visits although it is unclear why this would not be done for all follow-up visits if other measures of treatment outcome are being followed beyond 1 year. Also, I question whether evaluating CA19-9 3 weeks
--

	after SABR is useful as it is routine to see transient elevations in CA19-9 up to at least 4 weeks afterwards and thus assessing at 3 weeks may lead to values that are confounding and not meaningful. -should quality of life be a secondary endpoint given the emphasis on toxicity assessment in this study? -in the Background, data from Chuong et al are described from a prior meeting and they have since been published (PMID 35814383) -the criteria that define DLTs are limited and in my opinion should be expanded. Should grade 3+ toxicities including bowel/stomach ulcer, fistula, perforation be included? Grade 3+ abdominal pain? Any Grade 5 treatment-related toxicity? -delayed grade 3+ GI toxicity is a known risk especially of 1-fraction SABR based on previous non-MR guided data. However these will fall outside of the DLT definition of within 3 months of SABR. It is unclear how such toxicities will be evaluated and whether the study is designed to stop early should there be an unacceptably high rate. This is a critical aspect of this study design to clarify. -there is not mention of whether surgery or IRE are permitted after SABR which would significantly influence potential toxicities and thus affect the primary endpoint. Please clarify whether other local therapy is prohibited after SABR and if not how this will be accounted for when reporting outcomes. -please also clarify whether additional systemic therapy is controlled after SABR given that one of the second points is assessing the freedom from second line chemotherapy. It is routine to deliver maintenance chemotherapy after SABR and it is unclear if such chemotherapy is prohibited and rather chemotherapy is only permitted upon disease progression. Also please clarify if there are restrictions on which systemic therapy regimens are permitted.
--	---

REVIEWER	Takahashi, Hidenori Osaka International Cancer Institute, Surgery
REVIEW RETURNED	12-Jan-2023

GENERAL COMMENTS	The authors described a protocol of phase I study investigating the safety of hypofractionated adaptive radiotherapy using the MR Liniac in localized pancreatic cancer (Emerald-Pancreas trial). The manuscript was well-written, and figures and tables were appropriately created. The trial plan is well-conceived, and I would like to point out a couple of minor points to be addressed. 1. The manuscript is lacking several key dates regarding the trial schedule. Has this study already started? The study protocol needs to include the actual (or estimated) Study Start Date and the estimated Study Completion Date. 2. The sentence starting with "Early data from...." (Line 2, the 4th paragraph, Page 4) is lacking the number of cited reference.
---

REVIEWER	Chen, Kathryn T. Los Angeles County Harbor-UCLA Medical Center, Surgery
REVIEW RETURNED	16-Jan-2023

GENERAL COMMENTS	The study design is reasonable and well written. Several minor questions: 1. Who will make the assessment that the patient is unresectable? 2. The cut off of 15% as acceptable for DLT actually seems high, considering that many current series report less than 0-10% DLT for SBRT in pancreatic cancer. Can you comment on how you established 15%?
--

VERSION 1 – AUTHOR RESPONSE

Reviewer: 1

Dr. Michael D. Chuong, Miami Cancer Institute Comments to the Author:

The authors present a phase I study design of ablative MR-guided radiation therapy for pancreas cancers that are not operable that includes 1-, 3-, and 5-fraction regimens. Overall the manuscript is well-written and addresses a topic that is of significant clinical interest to patients and the medical community.

Please consider the following:

-there is very little difference in absolute fraction numbers between 1, 3, and 5. It would be helpful for the reader to understand better the rationale for studying these 3 fractionation schedules as opposed to only 1 and 5 fractions for example. If all 3 regimens are deemed safe then do the investigators plan to proceed with other studies incorporating all 3 regimens?

-a phase II study (NCT03621644) of 50 Gy in 5 fractions delivered on MRIdian for pancreas cancers was recently presented at ASTRO showing very little grade 3+ toxicity. Considering including mention of this study and citing it. Also, how do the authors view the significance of the current trial that also is evaluating 50 Gy in 5 fractions on MRIdian in the context of the favorable results from the aforementioned study?

This phase I study is evaluating hypofractionation beyond the 'standard' 5-f SABR. The main aim of the

study is to test the feasibility of 3-f and 1-f regimens. During development of trial design, consideration was given to testing only single fraction regimen – however it was felt to be 'putting all your eggs in one basket' approach, given previous challenges with single-fraction pancreas radiation on CT platform. Therefore, testing feasibility of both the 1-f as well as 3-f regimen was felt to be appropriate. At the time of developing EMERALD, SMART (NCT03621644) was ongoing and therefore it was felt reasonable to keep the 5-f arm for the moment – as (1) it would allow a pragmatic (crude) nonrandomised comparison of side effects (2) allow patients whose constraints cannot be met on the 1f

or 3f arms to still receive high dose 5f SABR and (3) for UK to gain experience in 50Gy/5f SABR (compared to standard dose of 35-40Gy/5f). The trial is designed such that the 1f and the 3f arms are prioritized – after DLT assessment period of the first 3 patients in each of 3f and the 1f arms, prioritized

expansion cohorts will allow 10 patients to enter each of these 2 arms, before 1:1:1 allocation to 1f:3f:5f begins. With the data from NCT03621644 now available, Trial Management Group will consider whether the 5f arm could be dropped altogether through protocol amendment.

-although this is a phase I study primarily evaluating toxicity, the follow-up period is relatively lengthy at over 1 year with secondary endpoints evaluating efficacy which seems more consistent with a phase II or III study design. Would it therefore be advantageous to be a bit more selective in patients enrolled based on CA19-9, receipt of prior chemotherapy, etc? Otherwise the long-term efficacy outcomes will be hard to interpret and as such shortening the follow-up period may be advantageous.

The primary DLT assessment window is 3 months (which is the primary end-point). A CT scan is also mandated at 3-months to assess response. Subsequent management is largely dictated by local oncologists, and telemed appointments will capture late toxicity and outcome data. We agree with the reviewer that a 1-year follow up was not absolutely necessary for a phase I trial, but it was felt that the 1-year follow up period will allow us to capture preliminary efficacy data and late RT sideeffects which could inform the design and statistics of future trials.

-CA19-9 is assessed at baseline and then only at 2 of the follow-up visits although it is unclear why

this would not be done for all follow-up visits if other measures of treatment outcome are being followed beyond 1 year. Also, I question whether evaluating CA19-9 3 weeks after SABR is useful as it

is routine to see transient elevations in CA19-9 up to at least 4 weeks afterwards and thus assessing at 3 weeks may lead to values that are confounding and not meaningful.

This is a pragmatic consideration, keeping in mind toxicity, not efficacy or biochemical response, is the primary end-point. It is a single-centre study so the ethics for doing specific tests is limited to Oxford. The recruitment catchment area for this study is national, and >80% of referrals are from outside the region. In the best interest of the patient, we have limited face to face follow up to two visits (week 3, ie 2 weeks after completion of treatment) and 3 months (response assessment and end of DLT period). At these visits we can mandate specific blood tests eg CA19.9. The rest of the follow ups are telemed and blood tests and other aspects of care are as per local oncologist's preferences, therefore it was not possible to mandate 12-weekly CA19.9s as this may not be the standard in the hospital they have come from

-should quality of life be a secondary endpoint given the emphasis on toxicity assessment in this study?

This being a phase 1 study, a QOL end-point was not considered, although we agree with reviewer that it would have been an interesting addition

-in the Background, data from Chuong et al are described from a prior meeting and they have since been published (PMID 35814383) -the criteria that define DLTs are limited and in my opinion should be expanded. Should grade 3+ toxicities including bowel/stomach ulcer, fistula, perforation be included? Grade 3+ abdominal pain? Any Grade 5 treatment-related toxicity?

-delayed grade 3+ GI toxicity is a known risk especially of 1-fraction SABR based on previous non-MR guided data. However these will fall outside of the DLT definition of within 3 months of SABR. It is unclear how such toxicities will be evaluated and whether the study is designed to stop early should there be an unacceptably high rate. This is a critical aspect of this study design to clarify.

Although the DLT period is 3 months, patients will undergo telemed follow up 3-monthly for 1 year and therefore toxicity data beyond 3 months will be collected. The safety committee which includes TMG members and 2 independent clinical oncologists with experience in pancreatic SABR will meet every time there is a DLT and to evaluate the toxicity within the trial.

-there is not mention of whether surgery or IRE are permitted after SABR which would significantly influence potential toxicities and thus affect the primary endpoint. Please clarify whether other local therapy is prohibited after SABR and if not how this will be accounted for when reporting outcomes.

-please also clarify whether additional systemic therapy is controlled after SABR given that one of the second points is assessing the freedom from second line chemotherapy. It is routine to deliver maintenance chemotherapy after SABR and it is unclear if such chemotherapy is prohibited and rather chemotherapy is only permitted upon disease progression. Also please clarify if there are restrictions on which systemic therapy regimens are permitted.

The inclusion criteria does specify unresectable/inoperable disease, therefore patient is unlikely to have surgery. UK has a nationally standardised approach and patients with borderline resectable disease are offered FOLFIRINOX or chemoradiation, not SABR.

IRE is not available within the National Health Service.

Reviewer: 2

Dr. Hidenori Takahashi, Osaka International Cancer Institute Comments to the Author:

The authors described a protocol of phase I study investigating the safety of hypofractionated adaptive radiotherapy using the MR Liniac in localized pancreatic cancer (Emerald-Pancreas trial). The manuscript was well-written, and figures and tables were appropriately created. The trial plan is well-conceived, and I would like to point out a couple of minor points to be addressed.

1. The manuscript is lacking several key dates regarding the trial schedule. Has this study already started? The study protocol needs to include the actual (or estimated) Study Start Date and the estimated Study Completion Date.

The manuscript has been updated to include these key dates (in background section)):

FPFV 25 Aug 2022, LPLV planned for 30 June 2024, end of study 31 Dec2024

2. The sentence starting with “Early data from....” (Line 2, the 4th paragraph, Page 4) is lacking the number of cited reference.

Amended

Reviewer: 3

Dr. Kathryn T. Chen, Los Angeles County Harbor-UCLA Medical Center Comments to the Author:

The study design is reasonable and well written. Several minor questions:

1. Who will make the assessment that the patient is unresectable?

As this is a phase 1 study with toxicity end-point, the entry criteria is broad and includes unresectable/resectable but medically inoperable/recurrent. The assessment of resectability is upto the referring oncologist. A central review of imaging of all patients in the trial is undertaken by an Oxford pancreatic radiologist to determine suitability for trial, and furthermore, the radiologist also reviews the oncologist-outlined tumour and OAR contours.

2. The cut off of 15% as acceptable for DLT actually seems high, considering that many current series report less than 0-10% DLT for SBRT in pancreatic cancer. Can you comment on how you established 15%?

This is a small phase I study with no more than 20 patients is expected to be recruited in each of the 3f and 1f arms. A go/no-go decision made on 2/20 (10%) encountering DLT was felt to be too few events with an element of chance and therefore a higher cut-off was chosen.

VERSION 2 – REVIEW

REVIEWER	Chuong, Michael D. Miami Cancer Institute Honoraria and research funding from ViewRay, Inc.
REVIEW RETURNED	10-Jul-2023

GENERAL COMMENTS	The authors have appropriately responded to my questions.
---

REVIEWER	Takahashi, Hidenori Osaka International Cancer Institute, Surgery
REVIEW RETURNED	15-Jul-2023

GENERAL COMMENTS	I have no additional comments.
--------------------------------